

# Optical bounds on many-electron localization

Ivo Souza[1,2], Richard M. Martin[3,4] and Massimiliano Stengel[5,6]

**1** Centro de Física de Materiales, Universidad del País Vasco, 20018 San Sebastián, Spain
**2** Ikerbasque Foundation, 48013 Bilbao, Spain
**3** Department of Physics, University of Illinois at Urbana-Champaign,
Urbana, Illinois 61801, USA
**4** Department of Applied Physics, Stanford University, Stanford, California 94305, USA
**5** Institut de Ciència de Materials de Barcelona (ICMAB-CSIC),
Campus UAB, 08193 Bellaterra, Spain
**6** ICREA-Institució Catalana de Recerca i Estudis Avançats, 08010 Barcelona, Spain

## Abstract

We establish rigorous inequalities between different electronic properties linked to optical sum rules, and organize them into weak and strong bounds on three characteristic properties of insulators: electron localization length $\ell$ (the quantum fluctuations in polarization), electric susceptibility $\chi$, and optical gap $E_\mathrm{G}$. All-electron and valence-only versions of the bounds are given, and the latter are found to be more informative. The bounds on $\ell$ are particularly interesting, as they provide reasonably tight estimates for an ellusive ground-state property – the average localization length of valence electrons – from tabulated experimental data: electron density, high-frequency dielectric constant, and optical gap. The localization lengths estimated in this way for several materials follow simple chemical trends, especially for the alkali halides. We also illustrate our findings via analytically solvable harmonic oscillator models, which reveal an intriguing connection to the physics of long-ranged van der Waals forces.



# 1   Introduction

The low-frequency electronic conductivity,

$$\sigma_{aa}(\omega) = \operatorname{Re}\sigma_{aa}(\omega) + i\operatorname{Im}\sigma_{aa}(\omega), \tag{1}$$

displays sharply different behaviors in metals and in insulators. To characterize those behaviors one may define

$$D_{aa} = \pi \lim_{\omega \to 0} \omega \operatorname{Im}\sigma_{aa}(\omega), \tag{2a}$$

$$\epsilon_0 \chi_{aa} = -\lim_{\omega \to 0} \omega^{-1} \operatorname{Im}\sigma_{aa}(\omega), \tag{2b}$$

where $\epsilon_0$ is the vacuum permittivity. The Drude weight $D_{aa}$ is finite in metals and vanishes in insulators, whereas the clamped-ion electric susceptibility $\chi_{aa}$ is finite in insulators and diverges in metals. The $1/\omega$ divergence of $\operatorname{Im}\sigma_{aa}(\omega)$ in perfect conductors is due to the acceleration of free electrons under an applied electric field, and its linear decrease with $\omega$ in insulators reflects the polarization of bound electrons in reaction to the field.

In 1964, Kohn proposed electron localization as the essential property of the insulating state, and showed that it leads directly to its distinctive electrical behavior [1]. He argued that the ground-state wave function $\Psi(\mathbf{r}_1, \ldots, \mathbf{r}_N)$ of an insulator in a periodic supercell breaks up into a sum of functions, $\Psi = \sum_M \Psi_M$, which are localized in disconnected regions of configuration space and have essentially vanishing overlap. Kohn went on to show that the disconectedness of $\Psi$ allows for the definition of an effective center-of-mass operator $\mathbf{X}/N$, even though the bare center-of-mass operator operator $(1/N)\sum_{i=1}^N \mathbf{r}_i$ is ill-defined under periodic boundary conditions. The operator $\mathbf{X}$ is based on sawtooth functions, whose discontinuities are placed in regions of configuration space where $\Psi$ becomes exponentially small [2].

The importance of $\mathbf{X}$ can be seen from the fact that its ground-state expectation value yields the electronic contribution to the macroscopic electric polarization ($\mathbf{P}$),

$$\mathbf{P}_{\mathrm{e}} = -|e|\langle\mathbf{X}\rangle/V, \tag{3}$$

where $V$ is the supercell volume. Thanks to the development of the modern theory of polarization, $\mathbf{P}$ is now understood as a fundamental bulk property of crystalline insulators, independent of surface termination modulo a discrete quantum of indeterminacy. In particular, within

a single-particle band picture, Eq. (3) reduces to a sum over the Wannier centers (Kohn's disconnected wave function pieces $\Psi_M$ can be viewed as "many-body Wannier functions"), or can be equivalently written as a Berry phase in momentum space [3,4]. Crucially, this theory asserts that bulk polarization is a property of the wave function and not of the charge density, in line with Kohn's view on electron localization.

In addition, Kohn's center-of-mass operator allows for the definition of an electron localization tensor [5]

$$\ell_{ab}^2 = \frac{1}{N} \left[ \langle X_a X_b \rangle - \langle X_a \rangle \langle X_b \rangle \right]. \tag{4}$$

The diagonal entries of this symmetric tensor carry the interpretation of a localization length squared, averaged over the total number of electrons, along the corresponding direction. In high-symmetry crystals, the localization tensor becomes isotropic:

$$\ell_{ab}^2 = \delta_{ab} \ell^2. \tag{5}$$

As in the case of $\mathbf{P}$, the localization tensor enjoys an elegant formulation in the framework of band theory, where it can be written as a quantum metric tensor [6] of the valence Bloch manifold [5,7,8], whose Cartesian trace is related to the Wannier spread [9]: see Appendix A. First-principles studies of the localization tensor have been carried out for tetrahedrally-coordinated semiconductors [10] and oxides [11].

Kohn did not directly relate the degree of wave function localization to any physical observable. An important step in that direction was taken shortly before the modern theory of polarization was developed. In Ref. [12], Kudinov proposed to quantify electron localization in insulators via the quantum fluctuations in the ground-state polarization [Eq. (4)], connecting them to the optical absorption spectrum by means of a fluctuation-dissipation relation. For a bulk crystal, such relation at zero temperature takes the form [5]

$$\ell_{ab}^2 = \frac{\hbar}{\pi e^2 n_e} \int_0^\infty d\omega \, \omega^{-1} \operatorname{Re} \sigma_{ab}^S(\omega), \tag{6}$$

where $n_e = N/V$ is the electron density, the superscript S denotes the symmetric part of the conductivity tensor, and the integral spans the positive-frequency optical absorption spectrum.[1] The trace of the localization tensor diverges in conductors by virtue of their nonzero DC Ohmic conductivity, while in insulators it remains finite.

The fluctuation-dissipation relation written above assumes a vanishing macroscopic electric field $\mathbf{E}$, as appropriate for transverse long-wave excitations. The needed generalization to accomodate more general electrical boundary conditions was given by Resta [13]. In particular, for longitudinal excitations where $\mathbf{E} = -\mathbf{P}/\epsilon_0$ ($\mathbf{D} = \mathbf{0}$), the fluctuation-dissipation relation becomes a sum rule for the energy-loss spectrum,

$$\tilde{\ell}^2(\hat{\mathbf{q}}) = \frac{\hbar \epsilon_0}{\pi e^2 n_e} \int_0^\infty d\omega \operatorname{Im} \left[ -\epsilon^{-1}(\hat{\mathbf{q}}, \omega) \right], \tag{7}$$

where the integrand is the (generally direction-dependent) $\mathbf{q} \to \mathbf{0}$ limit of the longitudinal inverse dielectric function. The quantum fluctuations encoded in the localization tensor depend on the electrical boundary conditions, and it is only under $\mathbf{E} = \mathbf{0}$ as assumed in Eq. (6) that its trace discriminates between insulators and metals [13]. In the following, we will deal mostly with the transverse localization tensor; when referring to longitudinal quantities, we will denote them with a tilde as done above.

---

[1]The conductivity tensor can be decomposed in three different ways: real and imaginary parts, $\operatorname{Re} \sigma$ and $\operatorname{Im} \sigma$; symmetric and antisymmetric parts, $\sigma^S$ and $\sigma^A$; Hermitian and anti-Hermitian parts, $\sigma^H$ and $\sigma^{AH}$. The Hermitian part of $\sigma$ (and hence $\operatorname{Re} \sigma^S$) is dissipative, while the anti-Hermitian part is reactive.

Table 1: Overview of the sum-rule inequalities on $\ell$ discussed in the present work. Those inequalities relate the electron localization length $\ell$ defined by Eqs. (4) and (5) to the optical gap $E_G$, the clamped-ion electric susceptibility $\chi$, and the electron density $n_e$. The last column contains equivalent energy relations involving the localization gap $E_L$ and the Penn gap $E_P$, which will be defined shortly [see Eq. (11)].

| Length relations | References | Comments | Energy relations |
|---|---|---|---|
| $\ell \leq \ell_{++}$ | [5] | $\ell_{++}^2 \propto 1/E_G$ | $E_L \geq E_G$ |
| | | Weak upper bound | |
| $\ell_- \leq \ell$ | [14,15] | $\ell_-^2 \propto \chi E_G/n_e$ | $E_P^2/E_G \geq E_L$ |
| | | Lower bound | |
| | | Sum-rule derivation in [15] | |
| $\ell \leq \ell_+$ | [16,17], | $\ell_+^2 \propto \sqrt{\chi/n_e}$ | $E_L \geq E_P$ |
| | this work | Strong upper bound | |
| $\ell_+ \leq \ell_{++}$ | This work | Equivalent to $\ell_- \leq \ell_+, \ell_{++}$ | $E_P \geq E_G$ |
| $\ell_- \leq \ell \leq \ell_+ \leq \ell_{++}$ | This work | Chained inequalities | $E_P^2/E_G \geq E_L \geq E_P \geq E_G$ |

Although Eq. (6) provides a way of extracting the transverse localization length $\ell$ from the optical absorption spectrum, we are not aware of any experimental work in that direction. As discussed in Ref. [15], an alternative is to estimate $\ell$ via rigorous upper and lower bounds involving readily-available experimental data: electron density $n_e$, clamped-ion electric susceptibility $\chi$, and minimum optical gap $E_G$ (see Table 1). This strategy was used recently to estimate $2\pi n_e \ell^2$ (the quantum metric of the filled bands) for a number of materials [17,18].

In this work, we employ a sum-rule approach to establish weak and strong bounds on $\ell$, $\chi$, and $E_G$. We give two formulations of the bounds – all electron and valence-only – and argue that the valence-only formulation, even if approximate, is more informative. This is confirmed by an explicit evaluation of the bounds on $\ell$ for a series of materials; the strong bound is found to be much tighter than the weak one, and the valence-only formulation reveals simple chemical trends. To illustrate the impact of long-ranged electrostatics on the polarization fluctuations [13], we apply our formalism to analytically solvable systems of harmonic oscillators. This exercise reveals an intriguing connection to the physics of van der Waals (dispersion) forces, and clarifies the central role of electron-electron correlation in the determination of the optical bounds.

The manuscript is organized as follows. In Sec. 2 the inverse moments of the optical absorption spectrum are introduced, the sum rules for the three leading moments are stated, and average optical gaps are defined. In Sec. 3, sum-rule inequalities are established for the inverse moments and for the average gaps; the latter are then organized into chained inequalities, from which various bounds on $\ell$, $\chi$, and $E_G$ are deduced. In Sec. 4 those bounds are examined for several exactly-solvable models, including harmonic oscillator models coupled by dispersion interactions. In Sec. 5 the localization length $\ell$ is estimated for several materials using the all-electron and valence-only varieties of the bounds, and the observed trends are discussed. We conclude in Sec. 6 with a summary, and provide some accessory results in three appendices.

## 2 Sum rules and average gaps

For light with linear polarization along direction $\hat{\mathbf{n}}$, we define the inverse moments of the positive-frequency optical absorption spectrum at zero temperature as

$$I_p(\hat{\mathbf{n}}) = \frac{2}{\pi} \int_0^\infty d\omega\, \omega^{-p} \operatorname{Re} \sigma_{ab}^{\mathrm{S}}(\omega)\, \hat{n}_a \hat{n}_b \,, \tag{8}$$

where $p \geq 0$, a summation over repeated Cartesian indices is implied, and the $2/\pi$ factor was included for convenience in writing the sum rules below. For simplicity we will assume cubic symmetry or higher so that $\sigma_{ab}^{\mathrm{S}} = \delta_{ab} \sigma^{\mathrm{S}}$, rendering $I_p$ independent of $\hat{\mathbf{n}}$,

$$I_p = \frac{2}{\pi} \int_0^\infty d\omega\, \omega^{-p} \operatorname{Re} \sigma^{\mathrm{S}}(\omega) \,. \tag{9}$$

The inverse spectral moments with $p = 0, 1, 2$ satisfy

$$I_0 = \frac{e^2 n_{\mathrm{e}}}{m_{\mathrm{e}}} \equiv \epsilon_0 \omega_{\mathrm{p}}^2 \,, \tag{10a}$$

$$I_1 = \frac{2e^2}{\hbar} n_{\mathrm{e}} \ell^2 \,, \tag{10b}$$

$$I_2 = \epsilon_0 \chi \equiv \epsilon_0 (\epsilon - 1) \,, \tag{10c}$$

where we have introduced the static electronic permittivity $\epsilon$ (often denoted as $\epsilon_\infty$), and the plasma frequency $\omega_{\mathrm{p}}$.[2] The above identities are respectively the oscillator-strength sum rule, the fluctuation-dissipation relation of Eq. (6) [with $\ell^2$ given by Eq. (5)], and the electric-susceptibility sum rule. All three sum rules converge for insulators, while in metals $I_1$ and $I_2$ diverge as a result of the nonzero DC conductivity. Equations (10a) and (10c) follow from the Kramers-Krönig relations, which in the case of (10a) must be combined with the observation that at sufficiently high frequencies the medium responds to an electromagnetic disturbance like a free-electron gas [20, 21]. The corresponding sum rules for atomic systems are well known [22–24]. In solid-state physics, atomic-like sum rules have been used to characterize F centers in alkali halide crystals [25]; in particular, from the ratio between the $I_1$ and $I_0$ moments of the F-center absorption band one can deduce, under the effective-mass approximation, its mean radius in the ground state.

To proceed, we find it useful to define a "localization gap" $E_{\mathrm{L}}$ and a "Penn gap" $E_{\mathrm{P}}$ as [15]

$$E_{\mathrm{L}}^{-1} = \frac{\hbar^{-1} I_1}{I_0} \,, \qquad E_{\mathrm{P}}^{-2} = \frac{\hbar^{-2} I_2}{I_0} \,. \tag{11}$$

These average inverse excitation energies weighted by the transition strength [24] will be denoted as (inverse) "average gaps." Using Eq. (10) and writing $\hbar^2/2m_{\mathrm{e}}$ as $a_0^2 \mathrm{Ry}$ ($a_0$ is the Bohr radius and Ry is the Rydberg unit of energy), we obtain

$$E_{\mathrm{L}} = \frac{\hbar^2}{2m_{\mathrm{e}} \ell^2} \iff \left(\frac{\ell}{a_0}\right)^2 = \frac{\mathrm{Ry}}{E_{\mathrm{L}}} \,, \tag{12}$$

---

[2]In general, the frequency $\omega_{\mathrm{p}}$ defined by Eq. (10a) does not correspond to a physical resonance of the medium. The physical meaning of the parameter $\omega_{\mathrm{p}}$ is provided by the free-electron-like behavior of the dielectric function at frequencies far above the deepest core-level resonance [19, 20]: $\epsilon(\omega)/\epsilon_0 \simeq 1 - \omega_{\mathrm{p}}^2/\omega^2$. In a real plasma, electrons are free and the range of validity of this formula is very broad, including $\omega < \omega_{\mathrm{p}}$ [19]. In that case, $\omega_{\mathrm{p}}$ does correspond to a physical resonance of the medium.

and

$$\chi = \left(\frac{\hbar\omega_{\mathrm{p}}}{E_{\mathrm{P}}}\right)^2, \tag{13}$$

the latter being the standard definition of the Penn gap in semiconductor physics [26, 27].

As shown below and already indicated in Table 1, the inequalities of interest can be expressed concisely as relations among three characteristic energy scales of the band structure: optical gap $E_{\mathrm{G}}$ (the energy threshold for optical absorption), Penn gap, and localization gap.

## 3 Sum-rule inequalities

If the unperturbed system is in thermodynamic equilibrium, we have [20]

$$\sigma^S(\omega) \geq 0, \quad \text{for } \omega > 0. \tag{14}$$

From this condition, one can readily obtain two types of inequalities involving different spectral moments [24]. The first type are of the form

$$I_{p+q} \leq \frac{\hbar}{E_{\mathrm{G}}} I_{p+q-1} \leq \dots \leq \left(\frac{\hbar}{E_{\mathrm{G}}}\right)^q I_p, \tag{15}$$

where $q > 0$; the second,

$$I_p^2 \leq I_{p-1}I_{p+1}, \tag{16}$$

follow from the Cauchy-Bunyakovsky-Schwarz inequality

$$\left(\int_0^\infty d\omega\, f(\omega)g(\omega)\right)^2 \leq \left(\int_0^\infty d\omega\, f(\omega)^2\right)\left(\int_0^\infty d\omega\, g(\omega)^2\right), \tag{17}$$

by setting $f(\omega) = \omega^{-(p-1)/2}\sqrt{\mathrm{Re}\,\sigma^S(\omega)}$ and $g(\omega) = \omega^{-(p+1)/2}\sqrt{\mathrm{Re}\,\sigma^S(\omega)}$. Both types of inequalities saturate in the limit of a narrow absorption spectrum concentrated at $E_{\mathrm{G}}$ [24].

The average gaps introduced in Eq. (11) satisfy

$$E_{\mathrm{L}} \geq E_{\mathrm{P}} \geq E_{\mathrm{G}}, \qquad E_{\mathrm{P}}^2 \geq E_{\mathrm{G}}E_{\mathrm{L}}, \tag{18}$$

with the relation $E_{\mathrm{L}} \geq E_{\mathrm{P}}$ coming from Eq. (16) and the others from Eq. (15); as expected, the average gaps $E_{\mathrm{L}}$ and $E_{\mathrm{P}}$ cannot be smaller than the minimum gap $E_{\mathrm{G}}$. Equation (18) allows to bracket $E_{\mathrm{L}}$ as $E_{\mathrm{P}}^2/E_{\mathrm{G}} \geq E_{\mathrm{L}} \geq E_{\mathrm{P}} \geq E_{\mathrm{G}}$ and $E_{\mathrm{P}}$ as $E_{\mathrm{L}}^2 \geq E_{\mathrm{P}}^2 \geq E_{\mathrm{G}}E_{\mathrm{L}} \geq E_{\mathrm{G}}^2$; combined with Eqs. (12) and (13), these chained inequalities yield

$$\frac{\epsilon_0 \chi E_{\mathrm{G}}}{2e^2 n_{\mathrm{e}}} \leq \ell^2 \leq \frac{\hbar}{2|e|}\sqrt{\frac{\epsilon_0 \chi}{m_{\mathrm{e}}n_{\mathrm{e}}}} \leq \frac{\hbar^2}{2m_{\mathrm{e}}E_{\mathrm{G}}} \qquad \left(\ell_-^2 \leq \ell^2 \leq \ell_+^2 \leq \ell_{++}^2\right), \tag{19a}$$

$$\frac{4e^2 m_{\mathrm{e}}n_{\mathrm{e}}\ell^4}{\epsilon_0 \hbar^2} \leq \chi \leq \frac{2e^2 n_{\mathrm{e}}\ell^2}{\epsilon_0 E_{\mathrm{G}}} \leq \frac{\hbar^2 e^2 n_{\mathrm{e}}}{\epsilon_0 m_{\mathrm{e}}E_{\mathrm{G}}^2}. \tag{19b}$$

We will refer to $\ell_-$ as the lower bound on $\ell$, and to $\ell_+$ and $\ell_{++}$ as the strong and weak upper bounds, respectively; the same terminology will be used for the bounds on $\chi$. The weak upper bounds on $\ell$ [5] and on $\chi$ [28] reflect the intuitive notion that wide-gap materials tend to have more localized and less polarizable electrons.

The bounds on $\ell$ are particularly interesting, as they only involve parameters that are tabulated for many materials: electron density, electric susceptibility, and optical gap. Since $\ell$ itself is not commonly measured, those bounds provide a simple and practical way of estimating its

value. Note that the weak upper bound $\ell_{++}$ only depends on the inverse minimum gap; this is a delicate quantity, especially for narrow-gap semiconductors, and it is not representative of the entire spectrum (the nature of the electron system can be very different for materials with the same minimum gap). The localization length is instead a global property of the electron system, and the value of $E_G$ is not its most relevant descriptor; for example, $\ell_{++}$ diverges in the same way for all materials as $E_G$ is tuned to zero. We therefore expect $\ell_{++}$ to give a relatively poor estimate for $\ell$ in real systems. The strong upper bound $\ell_+$ depends instead on $\chi$ and $n_e$ via the average Penn gap, which is much more representative of the entire spectrum. As for the lower bound $\ell_-$, it depends on both $E_G$ and $E_P$; there is still some dependence on the minimum gap, but it is a smaller effect than for $\ell_{++}$.

The relations in Eq. (18) can also be arranged as $E_G \leq E_P^2/E_L \leq E_P \leq E_L$ to place bounds on the optical gap,

$$E_G \leq \frac{2e^2 n_e \ell^2}{\epsilon_0 \chi} \leq \hbar |e| \sqrt{\frac{n_e}{m_e \epsilon_0 \chi}} \leq \frac{\hbar^2}{2 m_e \ell^2} \,. \tag{20}$$

It is significant that there are several upper bounds, but no lower bound. This is consistent with the existence of electronic systems without an energy gap that are strict insulators [1, 2].

Although our focus has been on transverse long-wave modes, similar results hold for longitudinal modes [17, 28]. The only changes to Eqs. (19) and (20) are (see Appendix B)

$$E_G \rightarrow \tilde{E}_G \,, \qquad \ell \rightarrow \tilde{\ell} \,, \qquad \chi \rightarrow 1 - \epsilon^{-1} \,, \tag{21}$$

where $\tilde{E}_G$ is the minimum energy for long-wave longitudinal excitations (plasmon gap), and $\tilde{\ell}$ was introduced in Eq. (7) (in high-symmetry crystals, $\tilde{\ell}$ does not dependend on $\hat{\mathbf{q}}$). The lower and strong upper bounds on $\tilde{\ell}^2$ are given in Ref. [17] (in terms of $2\pi n_e \tilde{\ell}^2$), and the weak upper bound on $1 - \epsilon^{-1}$ is given in Ref. [28].

In closing, we comment on the applicability of the above relations to Chern insultors (CIs). The general character of the sum rules in Eq. (10) suggests that the inequalities deduced from them remain valid for CIs. The subtlety is that CIs occupy a middle ground between metals and ordinary insulators [4], and the $I_1$ and $I_2$ sum rules diverge for metals. On the other hand, all three sum rules involve the symmetric (time even) part of the optical conductivity, whereas the distinction between ordinary and Chern insulators rests with the antisymmetric (time odd) part; from this we can conclude that the inequalities obtained above do apply to CIs, even if such materials fall outside the scope of Kohn's theory of the insulating state. Indeed, while the total Wannier spread diverges in a CI, its gauge invariant part proportional to $\ell^2$ remains finite [29], consistent with the weak upper bound on $\ell^2$. Likewise, the weak upper bound on $\chi$ implies that the susceptibility remains finite in CIs, even though the concept of spontaneous polarization requires special care [30].

# 4 Analytically solvable models

To build intuition on the bounds obtained above, we will now apply them to several models that can be treated analytically. For the first few examples dealing with finite systems, we introduce a polarizability per electron via the relation $\mathbf{d} = N\alpha\mathbf{E}_0$; here $\mathbf{d}$ is the dipole moment induced on the $N$-electron system by the applied electric field $\mathbf{E}_0$. To use the bulk relations (19) and (20), we place the system in a periodic supercell. In the limit where the supercell dimensions far exceed those of the system, the applied field $\mathbf{E}_0$ generates a macroscopic field $\mathbf{E} = \mathbf{E}_0$ in the effective medium; from $\mathbf{P} = \epsilon_0 \chi \mathbf{E} = \mathbf{d}/V$ we get $\chi = n_e \alpha/\epsilon_0 + \mathcal{O}(V^{-1})$, where the additional terms (originating from the Clausius-Mossotti relation, see Sec. 4.4) vanish in the assumed

limit of large $V$. Plugging this expression for $\chi$ into Eqs. (19) and (20) gives

$$\frac{\alpha E_{\text{G}}}{2e^2} \leq \ell^2 \leq \frac{\hbar}{2|e|}\sqrt{\frac{\alpha}{m_{\text{e}}}} \leq \frac{\hbar^2}{2m_{\text{e}}E_{\text{G}}}\,, \tag{22a}$$

$$\frac{4e^2 m_{\text{e}}\ell^4}{\hbar^2} \leq \alpha \leq \frac{2e^2\ell^2}{E_{\text{G}}} \leq \frac{\hbar^2 e^2}{m_{\text{e}}E_{\text{G}}^2}\,, \tag{22b}$$

$$E_{\text{G}} \leq \frac{2e^2\ell^2}{\alpha} \leq \frac{\hbar|e|}{\sqrt{m_{\text{e}}\alpha}} \leq \frac{\hbar^2}{2m_{\text{e}}\ell^2}\,. \tag{22c}$$

At this point we make contact with known results for atoms and molecules. The strong upper bound on $\alpha$, with $E_{\text{G}}$ replaced by a mean excitation energy $\Delta E$ and $3e^2\ell^2$ expressed as the dipole fluctuation $\langle d^2\rangle - \langle d\rangle^2$, becomes

$$\alpha \approx \frac{2}{3}\frac{\langle d^2\rangle - \langle d\rangle^2}{\Delta E}\,. \tag{23}$$

This estimate for the polarizability is discussed in Ref. [31], along with its relation to the fluctuation-dissipation relation. That textbook also gives an estimate for $\alpha$ in terms of the weak upper bound in Eq. (22b), invoking the oscillator-strength sum rule.

## 4.1 Hydrogen atom

Introducing the polarizability volume $\alpha' = \alpha/4\pi\epsilon_0$ [31], Eq. (22) becomes

$$\frac{1}{4}\frac{\alpha'}{a_0^3}\frac{E_{\text{G}}}{\text{Ry}} \leq \frac{\ell^2}{a_0^2} \leq \frac{1}{2}\sqrt{\frac{\alpha'}{a_0^3}} \leq \frac{\text{Ry}}{E_{\text{G}}}\,, \tag{24a}$$

$$4\frac{\ell^4}{a_0^4} \leq \frac{\alpha'}{a_0^3} \leq 4\frac{\text{Ry}}{E_{\text{G}}}\frac{\ell^2}{a_0^2} \leq 4\frac{\text{Ry}^2}{E_{\text{G}}^2}\,, \tag{24b}$$

$$\frac{E_{\text{G}}}{\text{Ry}} \leq 4\frac{a_0\ell^2}{\alpha'} \leq 2\sqrt{\frac{a_0^3}{\alpha'}} \leq \frac{a_0^2}{\ell^2}\,, \tag{24c}$$

where every fraction is dimensionless. For the nonrelativistic hydrogen atom we have [32,33]

$$E_{\text{G}} = 0.75\,\text{Ry}\,, \qquad \alpha' = 4.5 a_0^3\,, \qquad \ell^2 = a_0^2\,, \tag{25}$$

which plugged into Eq. (24) gives

$$\frac{27}{32} \leq \frac{\ell^2}{a_0^2} = 1 \leq \sqrt{\frac{9}{8}} \leq \frac{4}{3}\,, \tag{26a}$$

$$4 \leq \frac{\alpha'}{a_0^3} = 4.5 \leq \frac{16}{3} \leq \left(\frac{8}{3}\right)^2\,, \tag{26b}$$

$$\frac{E_{\text{G}}}{\text{Ry}} = 0.75 \leq \frac{8}{9} \leq \frac{2}{\sqrt{4.5}} \leq 1\,. \tag{26c}$$

The lower and strong upper bounds on $\alpha'$ are given in Ref. [24], and the latter is also discussed in Ref. [32] and in other textbooks. Interestingly, both bounds can be improved by means of correction terms involving positive moments of the absorption spectrum [24].

Taking the average of the lower and strong upper bounds on $\ell^2$ and on $\alpha'$ produces the reasonably accurate estimates $\ell^2 \approx 0.952 a_0^2$ and $\alpha' \approx 4.(6) a_0^3$. The estimates $\ell^2 \approx 1.089 a_0^2$

and $\alpha' \approx 5.(5)a_0^3$ obtained by taking the average of the lower and weak upper bounds are much less accurate, especially for $\alpha'$. We also note that the strong upper bound $\ell_+$ is closer to $\ell$ than the lower bound $\ell_-$. This supports the notion that $\ell_+$, being based solely on the average Penn gap, is more representative of the entire absorption spectrum than $\ell_-$, which also depends on the minimum gap. Further evidence that $\ell_+$ tends to track $\ell$ more closely than $\ell_-$ will be presented in Sec. 5 for crystalline materials.

## 4.2 Isotropic harmonic oscillator

For an electron trapped in an isotropic harmonic potential of frequency $\omega_0$ the parameters are [31, 32]

$$E_G = \hbar\omega_0, \quad \alpha = \frac{e^2}{m_e \omega_0^2} \equiv \alpha_0, \quad \ell^2 = \frac{\hbar}{2m_e \omega_0} \equiv \ell_0^2, \tag{27}$$

saturating all the inequalities in Eq. (22). This can be understood from the selection rules for the harmonic oscillator: as the only allowed dipole transition from the ground state is to the first excited state, the entire spectral weight is at $E_G$, producing the saturation.

## 4.3 Van der Waals dimer model

So far we have only discussed one-electron systems. To analyze the effect of electron correlations, we now consider a system of two identical harmonic oscillators 1 and 2 separated by $\mathbf{R}$. We think of these oscillators as vibrating electrical dipoles in which the $+e$ charges (ions) are held in the position of equilibrium while the $-e$ charges (electrons) vibrate about these equilibrium positions, their displacements being $\mathbf{r}_1$ and $\mathbf{r}_2$. In the limit where $r_1, r_2 \ll R$, this provides a simple model for the van der Waals interaction [31, 34].

The interaction term is

$$H_{12} = \frac{e^2}{4\pi\epsilon_0} \left( \frac{1}{R} + \frac{1}{|\mathbf{R} + \mathbf{r}_1 - \mathbf{r}_2|} - \frac{1}{|\mathbf{R} + \mathbf{r}_1|} - \frac{1}{|\mathbf{R} - \mathbf{r}_2|} \right). \tag{28}$$

In the approximation $r_1, r_2 \ll R$ we expand Eq. (28) to obtain in lowest order

$$H_{12} \simeq \frac{e^2}{4\pi\epsilon_0} r_{1a} r_{2b} \left( \frac{\delta_{ab}}{R^3} - 3\frac{R_a R_b}{R^5} \right), \tag{29}$$

which is in the form of a dipole-dipole interaction. Orienting the Cartesian frame such that $\mathbf{R} = R\hat{\mathbf{x}}$ leads to

$$H_{12} \simeq \frac{e^2}{4\pi\epsilon_0} \left( -\frac{2}{R^3} x_1 x_2 + \frac{1}{R^3} y_1 y_2 + \frac{1}{R^3} z_1 z_2 \right)$$
$$\equiv H_{12}^{\parallel} + H_{12}^{\perp}, \tag{30}$$

where $H_{12}^{\parallel}$ denotes the first term and $H_{12}^{\perp}$ the other two.

For oscillations along $\mathbf{R}$ the only surviving term in Eq. (30) is $H_{12}^{\parallel}$, and we recover the 1D model of Ref. [34]. Denoting by $H_0$ the Hamiltonian of the two uncoupled oscillators, $H_0 + H_{12}^{\parallel}$ is diagonalized by the transformation

$$x_{\pm} = \frac{1}{\sqrt{2}} (x_1 \pm x_2), \tag{31}$$

together with a similar transformation for the momenta, resulting in two decoupled oscillators with frequencies

$$\omega_{\pm}^{\parallel} = \omega_0 \sqrt{1 \mp \frac{2\alpha_0'}{R^3}}, \tag{32}$$

where $\alpha'_0$ is the polarizability volume of a single oscillator.

For unrestricted 3D oscillations, the interaction term is given by the full Eq. (30). Now instead of two modes we have six modes. By following through the same derivation, we can split also the $y$ and $z$ modes into symmetric and antisymmetric combinations with frequencies

$$\omega^\perp_\pm = \omega_0 \sqrt{1 \pm \frac{\alpha'_0}{R^3}}\,, \tag{33}$$

thus for transverse oscillations the symmetric modes have higher frequency than the antisymmetric ones.

In the 3D model the parameters $E_G$, $\alpha$, and $\ell^2$ are anisotropic, carrying labels $\parallel$ or $\perp$. To evaluate the $\parallel$ components, note that the interaction with a field $\mathbf{E}_\parallel = E\hat{\mathbf{x}}$ is described by $eE(x_1 + x_2) = \sqrt{2}eEx_+$, and that $\ell^2_\parallel$ is defined via (4) in terms of $X \equiv x_1 + x_2 = \sqrt{2}x_+$. This means that only the symmetric mode participates, and with a simple calculation one finds that the three parameters are obtained by replacing $\omega_0$ with $\omega^\parallel_+$ in Eq. (27),

$$E^\parallel_G = \hbar\omega^\parallel_+ \simeq \hbar\omega_0\left(1 - \alpha'_0/R^3\right), \tag{34a}$$

$$\alpha_\parallel = \frac{e^2}{m_e\left(\omega^\parallel_+\right)^2} \simeq \alpha_0\left(1 + 2\alpha'_0/R^3\right), \tag{34b}$$

$$\ell^2_\parallel = \frac{\hbar}{2m_e\omega^\parallel_+} \simeq \ell^2_0\left(1 + \alpha'_0/R^3\right). \tag{34c}$$

The $\perp$ components are obtained by sending $\omega^\parallel_+ \to \omega^\perp_+$ and $\alpha'_0 \to -\alpha'_0/2$ in these expressions.

In conclusion, the van der Waals interaction reduces the optical gap and increases both the polarizability and the localization length in the axial direction of the dimer, and the opposite happens in the perpendicular directions. As the antisymmetric modes are dipole inactive, the entire spectral weight for light polarized along $\mathbf{R}$ or perpendicularly to it is concentrated at a single frequency $\hbar\omega^\parallel_+$ or $\hbar\omega^\perp_+$, respectively. In both cases the bounds in Eq. (22) remain saturated, just like for a single oscillator.

We emphasize that the explicit treatment of electron correlations is essential to obtain a qualitatively correct physical picture. For example, it is easy to show that the fluctuation-dissipation sum rule fails if the electron-electron interaction is treated at the mean-field level, e.g., within Hartree-Fock theory. Within Hartree-Fock, the dielectric susceptibility of the system of interacting oscillators is described exactly; nonetheless, the localization length is unaffected by the interaction and corresponds to that of the isolated monomer. This implies that the correct description of the macroscopic polarization fluctuations goes hand in hand with the ability of the theory to capture dispersion interactions between isolated bodies.

## 4.4 Van der Waals crystal model

As an extension of the dimer model, we now consider a periodic array of oscillators coupled by dipole-dipole interactions. The potential energy reads

$$U = \frac{1}{2}m_e\omega_0^2 \sum_{\mathbf{R}} \left|\mathbf{r}^{\mathbf{R}}\right|^2 + \frac{e^2}{4\pi\epsilon_0} \sum_{\mathbf{R}} \sum_{\mathbf{R}'\neq\mathbf{0}} \frac{r_a^{\mathbf{R}} r_b^{\mathbf{R}+\mathbf{R}'}}{2}\left(\frac{\delta_{ab}}{R'^3} - 3\frac{R'_a R'_b}{R'^5}\right), \tag{35}$$

where $\mathbf{r}^{\mathbf{R}}$ denotes the displacement of an electron away from its equilibrium position $\mathbf{R}$, taken to be a point on a Bravais lattice. A similar model was discussed in Ref. [28]; the only difference is that the positive charges, instead of being point charges placed at the lattice points, are smeared into a uniform background.

### 4.4.1 Dynamical matrix

As in the dimer model, the electrons are assumed to be strongly localized in the sense that the quantum fluctuations are small compared to the separation between the ions. The resulting potential bears many similarities to the form that appears in the context of lattice vibrations; we will therefore borrow the same terminology in discussing the relevant contributions to the electronic Hamiltonian.

To determine the normal modes of the system we first evaluate the force-constant matrix

$$D_{a\mathbf{R},b\mathbf{R}'} \equiv \frac{\partial^2 U}{\partial r_a^{\mathbf{R}} \partial r_b^{\mathbf{R}'}} = D_{a\mathbf{0},b\mathbf{R}'-\mathbf{R}}, \tag{36}$$

to find

$$D_{a\mathbf{0},b\mathbf{R}} = m_{\mathrm{e}}\omega_0^2 \delta_{ab}\delta_{\mathbf{R}\mathbf{0}} + \frac{e^2}{4\pi\epsilon_0}(1-\delta_{\mathbf{R}\mathbf{0}})\left(\frac{\delta_{ab}}{R^3} - 3\frac{R_a R_b}{R^5}\right), \tag{37}$$

and then convert it into a dynamical matrix using

$$D_{ab}(\mathbf{q}) = \frac{1}{m_{\mathrm{e}}}\sum_{\mathbf{R}} D_{a\mathbf{0},b\mathbf{R}} e^{-i\mathbf{q}\cdot\mathbf{R}}. \tag{38}$$

The result is

$$D_{ab}(\mathbf{q}) = \omega_0^2\delta_{ab} + C_{ab}(\mathbf{q}), \tag{39}$$

where

$$C_{ab}(\mathbf{q}) = \frac{e^2/m_{\mathrm{e}}}{4\pi\epsilon_0}\sum_{\mathbf{R}\neq\mathbf{0}} e^{-i\mathbf{q}\cdot\mathbf{R}}\left(\frac{\delta_{ab}}{R^3} - 3\frac{R_a R_b}{R^5}\right). \tag{40}$$

To carry out the above lattice sum it is convenient to work in reciprocal space, where the interaction can be recast as a rapidly converging Ewald summation,

$$C_{ab}(\mathbf{q}) = \frac{e^2/m_{\mathrm{e}}}{4\pi\epsilon_0}\left(\frac{4\pi}{\Omega}\sum_{\mathbf{G}}' \frac{K_a K_b}{K^2} e^{-\frac{K^2\sigma^2}{4}} - \delta_{ab}\frac{4}{3\sqrt{\pi}\sigma^3}\right), \qquad \mathbf{K} = \mathbf{G} + \mathbf{q}, \tag{41}$$

with $\Omega$ the volume of a primitive cell. The primed sum excludes the divergent $\mathbf{G} + \mathbf{q} = \mathbf{0}$ term, and the second term removes the self-interaction of the dipole in the origin cell; the result is independent of the Ewald parameter $\sigma$ provided that $\sigma \ll R$ for all $\mathbf{R} \neq \mathbf{0}$.

By diagonalizing the $3\times 3$ matrix $C(\mathbf{q})$ at every point in the Brillouin zone, we have rewritten the problem as a set of independent oscillators. In particular, we have three modes at each $\mathbf{q}$ with frequencies

$$\omega_i^2(\mathbf{q}) = \omega_0^2 + \lambda_i(\mathbf{q}), \tag{42}$$

where $\lambda_i(\mathbf{q})$ are the eigenvalues of $C(\mathbf{q})$.

In Appendix C, we calculate the zero-point energy of this model by collecting the contributions from all normal modes across the Brillouin zone.

### 4.4.2 Long-wave limit

The $\mathbf{q} \to \mathbf{0}$ limit is particularly relevant to our discussion, since it corresponds to the collective displacement of the electronic center of mass. For a cubic lattice we find two TO modes and one LO mode where

$$\lambda_{\mathrm{TO}} = -\frac{1}{3}\omega_{\mathrm{p}}^2, \qquad \lambda_{\mathrm{LO}} = \frac{2}{3}\omega_{\mathrm{p}}^2, \tag{43}$$

with $\omega_{\mathrm{p}}^2 = e^2/\epsilon_0 m_{\mathrm{e}}\Omega$. To obtain this result note that the matrix $C(\mathbf{q})$ is traceless so that $2\lambda_{\mathrm{TO}} + \lambda_{\mathrm{LO}} = 0$, and that $\lambda_{\mathrm{LO}} = \lambda_{\mathrm{TO}} + \omega_{\mathrm{p}}^2$, where $\omega_{\mathrm{p}}^2$ is the contribution from the $\mathbf{G} = \mathbf{0}$ term in Eq. (41) when $\mathbf{q} \to \mathbf{0}$.

The dielectric susceptibility and permittivity are readily given in terms of the TO mode frequency,

$$\chi = \frac{\omega_{\text{p}}^2}{\omega_{\text{TO}}^2}, \qquad \epsilon = 1 + \chi. \tag{44}$$

Then, based on the above, we can quickly verify that the following results hold,

$$\epsilon = \frac{\omega_{\text{LO}}^2}{\omega_{\text{TO}}^2}, \qquad \frac{\epsilon - 1}{\epsilon + 2} = \frac{\alpha_0}{3\epsilon_0 \Omega}. \tag{45}$$

The first result is the Lyddane-Sachs-Teller relation [34], valid for a single-mode dielectric. The second is the Clausius-Mossotti relation [34], linking the macroscopic permittivity to the molecular polarizability $\alpha_0$.

For the TO modes we have

$$E_{\text{G}} = \hbar\omega_{\text{TO}}, \qquad \chi = \frac{\omega_{\text{p}}^2}{\omega_{\text{TO}}^2}, \qquad \ell^2 = \frac{\hbar}{2m_{\text{e}}\omega_{\text{TO}}}. \tag{46}$$

When plugged into Eqs. (12) and (13) these parameters give $E_{\text{L}} = E_{\text{P}} = E_{\text{G}}$, saturating all the bounds in Eqs. (19) and (20). The parameters for the LO modes are

$$\tilde{E}_{\text{G}} = \hbar\omega_{\text{LO}}, \qquad 1 - \epsilon^{-1} = \frac{\omega_{\text{p}}^2}{\omega_{\text{LO}}^2}, \qquad \tilde{\ell}^2 = \frac{\hbar}{2m_{\text{e}}\omega_{\text{LO}}}, \tag{47}$$

and again the corresponding bounds, obtained by modifying Eqs. (19) and (20) according to Eq. (21), are saturated.

## 5 Real materials

Starting from experimental data, we have evaluated the bounds on $\ell$ in Eq. (19a) for a number of materials. To visualize the results, it is helpful to bring that equation to the form

$$\frac{\text{Ry}}{E_{\text{P}}}\sqrt{\frac{E_{\text{G}}}{\text{Ry}}} \leq \frac{\ell}{a_0} \leq \sqrt{\frac{\text{Ry}}{E_{\text{P}}}} \leq \sqrt{\frac{\text{Ry}}{E_{\text{G}}}}, \tag{48}$$

which suggests plotting the data as shown schematically in Fig. 1. Given a data point (large blue dot), the range $[\ell_-, \ell_+]$ in units of $a_0$ is obtained by drawing horizontal and vertical line segments from it to the diagonal dashed line; its projection on that line (small black dot) yields

$$\ell \approx (\ell_+ + \ell_-)/2, \tag{49}$$

which we will refer to as the "strong bound" estimate, as opposed to the "weak bound" estimate obtained by replacing $\ell_+$ with $\ell_{++}$ in the expression above.

In the following, we use Eqs. (48) and (49) to estimate the electron localization length in different classes of materials. The needed experimental data are the optical gap (the lowest energy for optical absorption), the electron density, and the clamped-ion electric susceptibility; the last two enter via Eq. (13) for the Penn gap.

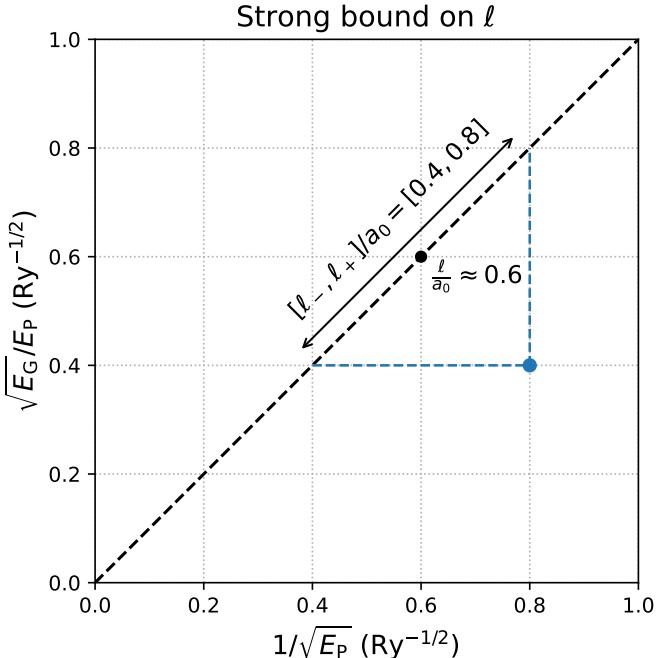

Figure 1: Schematic representation of the strong bound on $\ell$ in Eq. (48). For the weak bound, replace $E_P \to E_G$ on the horizontal axis; since $E_G \leq E_P$, the data point (blue dot) will move to the right, resulting in a wider range $[\ell_-, \ell_{++}]$.

## 5.1 Rocksalt alkali halides

Figure 2 shows the results obtained for alkali halides with the rocksalt structure. Consider first the top panels, where $E_P$ was calculated from the total electron density $n_e$ including inner core electrons. Such "all-electron" bounds inevitably provide average localization lengths that include those tight inert states; as a consequence, the bounds are rather loose not only on the left panel (weak bound) but also on the right panel (strong bound). On the left panel the upper bound is independent of $n_e$, and hence it is insensitive to the different localization lengths of valence and core electrons. This is not the case for the right panel, where the tighter upper bound containing $n_e$ narrows down the estimates for $\ell$; nevertheless, the data points are still quite far from the diagonal.

To rationalize the results for the strong bound, note that

$$\frac{\ell_+ - \ell_-}{a_0} = \sqrt{\frac{Ry}{E_P}} \left(1 - \sqrt{\frac{E_G}{E_P}}\right), \tag{50}$$

and thus the range $[\ell_-, \ell_+]$ gets tighter and tighter as $E_P$ gets closer to $E_G$. Since $E_P \propto \sqrt{n_e}$, the inclusion of core electrons goes in the opposite direction, and the range $[\ell_-, \ell_+]$ tends to increase as we move down the periodic table. This can be seen in the top-right panel of Fig. 2, where the distance from the diagonal line increases from the fluorides to the chlorides, from these to the bromides, and from these to the iodides.

It would be much more relevant for physical properties if one could estimate the average localization length of the valence electrons only. Here, we take the simple approach of replacing $E_P$ in Eq. (48) with a valence Penn gap calculated from the valence electron density.[3] The bounds on $\ell$ obtained in this manner are presented in the bottom panels of Fig. 2. As a result

---

[3]Such replacement assumes that valence-only versions of the sum rules in Eq. (10) can be formulated, which requires the excitation energies of core electrons to be well separated in energy from those of valence electrons [21].

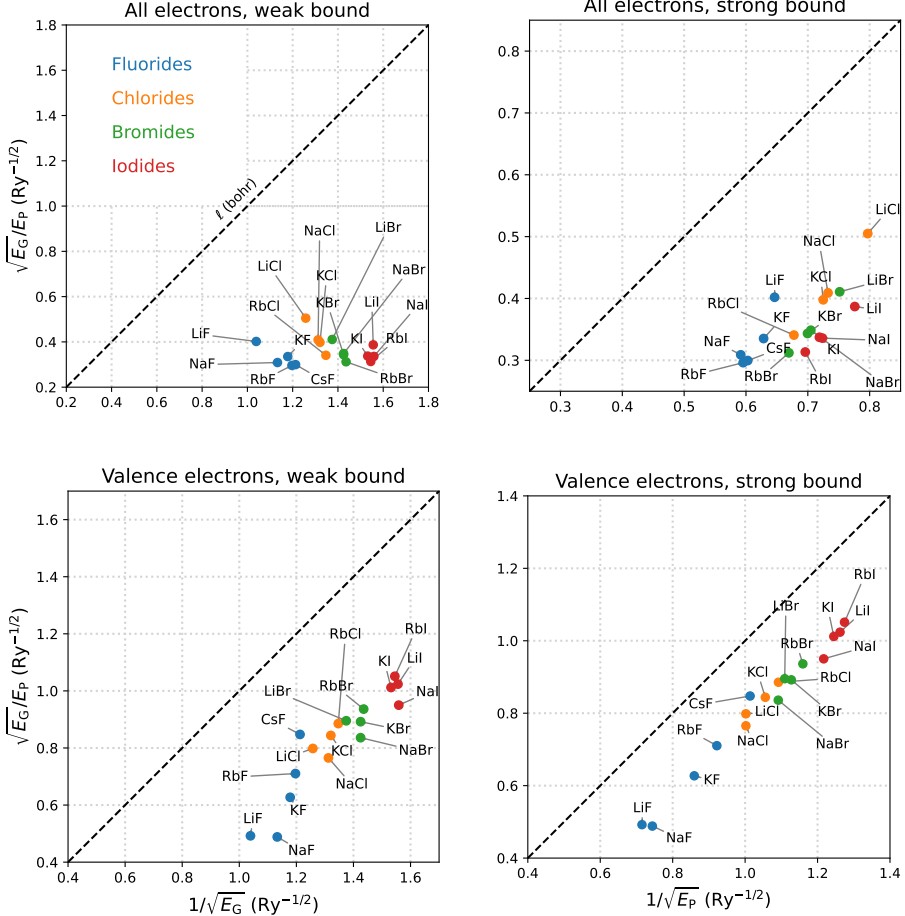

Figure 2: Bounds on $\ell$ for the rocksalt alkali halides, plotted using the scheme outlined in Fig. 1. The weak and strong bounds are represented on the left and right panels, respectively, while the top and bottom panels show all-electron and valence-electron results, respectively, with $E_P$ defined accordingly in each case.

of discarding the core electrons the data points move closer to diagonal line (the bounds get tighter), and their projections on that line move further up (the average localization lengths increase). Most interestingly, simple trends emerge in this valence-only formulation, with $\ell$ increasing from the lighter to the heavier halogens; this agrees with the intuition on chemical bonding in strongly ionic crystals [35]. The trend is most visible in the bottom right panel displaying the strong bound. The valence-only values for $\ell_-$, $\ell_+$, and $\ell_{++}$ are compiled in Table 2.

## 5.2 Tetrahedrally-coordinated materials

Figure 3 and Table 3 show the valence-only results obtained for materials with the diamond or the zincblende structure from groups IV, III-V, and II-VI in the periodic table. The trends are not as uniform as in the case of the halides because there is a larger range of gaps and susceptibilities. Nevertheless, one observes that the values of $\ell$ estimated from Eq. (49) tend to decrease with increasing ionicity, e.g., along the isoelectronic series Si → AlP and Ge → GaAs → ZnSe, as also found in Ref. [10]; this is consistent with the intuition that ionic bonding yields more localized electrons than covalent bonding. Accordingly, the estimated localization lengths in Table 3 tend to be larger than those in Table 2 for the strongly ionic alkali halides.

Table 2: Bounds on $\ell$ for the valence electrons in rocksalt alkali halides, estimated from experimental data: lattice constant $a$ (the electron density is $n_e = 32/a^3$, corresponding to eight valence electrons per formula unit), electronic permittivity $\epsilon$, and optical gap $E_G$. The values for $a$ and $\epsilon$ are from Ref. [35], and those for $E_G$ correspond to the lowest absorption peaks in Ref. [36]; the exceptions are LiF and LiI, for which $E_G$ are the excitonic gaps reported in Refs. [37] and [38], respectively.

| Crystal | $a$ (Å) | $\epsilon$ | $E_G$ (eV) | $\ell_-$ ($a_0$) | $\ell_+$ ($a_0$) | $\ell_{++}$ ($a_0$) |
|---|---|---|---|---|---|---|
| LiF | 4.02 | 1.96 | 12.6 | 0.40 | 0.65 | 1.04 |
| NaF | 4.62 | 1.74 | 10.6 | 0.31 | 0.59 | 1.13 |
| KF | 5.35 | 1.85 | 9.8 | 0.34 | 0.63 | 1.18 |
| RbF | 5.64 | 1.96 | 9.5 | 0.30 | 0.60 | 1.20 |
| CsF | 6.01 | 2.16 | 9.25 | 0.30 | 0.60 | 1.21 |
| LiCl | 5.13 | 2.78 | 8.6 | 0.50 | 0.80 | 1.26 |
| NaCl | 5.64 | 2.34 | 7.9 | 0.41 | 0.73 | 1.31 |
| KCl | 6.29 | 2.19 | 7.8 | 0.40 | 0.72 | 1.32 |
| RbCl | 6.58 | 2.19 | 7.5 | 0.34 | 0.68 | 1.35 |
| LiBr | 5.50 | 3.17 | 7.2 | 0.41 | 0.75 | 1.37 |
| NaBr | 5.97 | 2.59 | 6.7 | 0.35 | 0.70 | 1.43 |
| KBr | 6.60 | 2.34 | 6.7 | 0.34 | 0.70 | 1.43 |
| RbBr | 6.58 | 2.19 | 7.5 | 0.31 | 0.67 | 1.44 |
| LiI | 6.00 | 3.80 | 5.62 | 0.39 | 0.78 | 1.56 |
| NaI | 6.47 | 2.93 | 5.6 | 0.34 | 0.72 | 1.56 |
| KI | 7.07 | 2.62 | 5.8 | 0.34 | 0.72 | 1.53 |
| RbI | 7.34 | 2.59 | 5.7 | 0.31 | 0.70 | 1.54 |

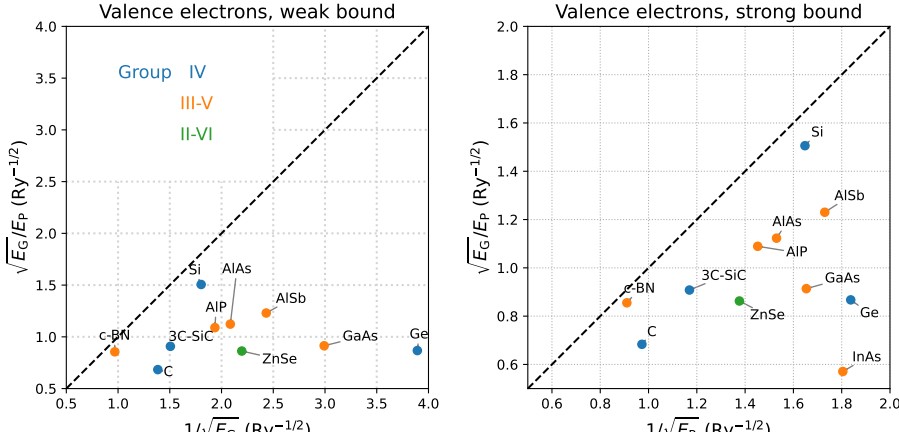

Figure 3: Same as the bottom two panels of Fig. 2, but for materials with the diamond or the zincblende structure. On the left panel, the data point for InAs is out of bounds.

How well do the $\ell$ values estimated from experimental data via Eq. (49) compare with those obtained from first-principles calculations? To address this question, in Fig. 4 we compare them with the *ab initio* values reported in Ref. [10]. The correlation is quite satisfactory, although the theoretical values tend to be somewhat larger. To explain this trend, one could invoke the band gap underestimation in density functional theory, which may well lead to a systematic overestimation of the calculated localization lengths. Since expressing $\ell$ as the average of $\ell_-$ and $\ell_+$ is an approximation, however, it is difficult to draw definitive conclusions,

Table 3: Same as Table 2, but for materials with the diamond or the zincblende structure. We assume four valence electrons per atom on average, so that $n_\mathrm{e} = 32/a^3$. The experimental data is from Ref. [39], where $E_\mathrm{G}$ is the direct gap.

| Crystal | $a\,(\text{Å})$ | $\epsilon$ | $E_\mathrm{G}$ (eV) | $\ell_-(a_0)$ | $\ell_+(a_0)$ | $\ell_{++}(a_0)$ |
|---|---|---|---|---|---|---|
| C | 3.57 | 5.7 | 7.1 | 0.68 | 0.97 | 1.38 |
| Si | 5.43 | 11.97 | 4.19 | 1.51 | 1.65 | 1.80 |
| Ge | 5.66 | 16.00 | 0.90 | 0.87 | 1.84 | 3.89 |
| 3C-SiC | 4.36 | 6.38 | 6.0 | 0.91 | 1.17 | 1.51 |
| c-BN | 3.62 | 4.46 | 14.5 | 0.86 | 0.91 | 0.97 |
| AlP | 5.46 | 7.5 | 3.63 | 1.09 | 1.45 | 1.94 |
| AlAs | 5.66 | 8.2 | 3.13 | 1.12 | 1.53 | 2.08 |
| AlSb | 6.14 | 10.24 | 2.3 | 1.23 | 1.73 | 2.43 |
| GaAs | 5.65 | 10.86 | 1.52 | 0.91 | 1.65 | 2.99 |
| InAs | 6.06 | 12.37 | 0.42 | 0.57 | 1.80 | 5.71 |
| ZnSe | 5.68 | 5.7 | 2.82 | 0.86 | 1.38 | 2.20 |

especially for cases like Ge where $\ell_-$ and $\ell_+$ are rather different. Yet, it is interesting to note that the upper bounds in Fig. 4 essentially fall on the diagonal in all cases, which means that they closely match the available theoretical data. (The lower bounds, involving the minimum gap, display a much larger scatter.) This gives further credit to our earlier statements that $\ell_+$ is a more robust indicator of the polarization fluctuation amplitude compared to $\ell_-$.

## 6 Conclusions

The use of sum-rule inequalities to estimate the electronic polarizability is well established in atomic and molecular physics [24,31]. The extension of those ideas to crystals and to other physical properties is not equally developed, and the results are scattered in the literature. In this work, we provided a unified perspective on several sum-rule inequalities for bulk systems, and organized them into chained inequalities providing bounds on three electronic properties of insulators: localization length $\ell$, static susceptibility $\chi$, and optical gap $E_\mathrm{G}$. As they are based on exact sum rules, those inequalities remain valid for correlated, disordered, and topological insulators, and in the presence of relativistic effects including spin-orbit coupling. The extension to low-symmetry crystals with anisotropic localization and susceptibility tensors is also straightforward. As an application, we estimated $\ell$ (a ground-state property) from readily available experimental data on the response properties $\chi$ and $E_\mathrm{G}$, together with the electron density. By focusing on the valence electrons, we obtained meaningful estimates for their average localization length that follow simple chemical trends.

The study of several exactly solvable models, from the hydrogen atom to isolated and coupled oscillators, provided useful insights. In particular, the coupled oscillator models illustrated how the fluctuation-dissipation relation breaks down at the mean-field level and critically requires an explicit treatment of dynamical correlations.



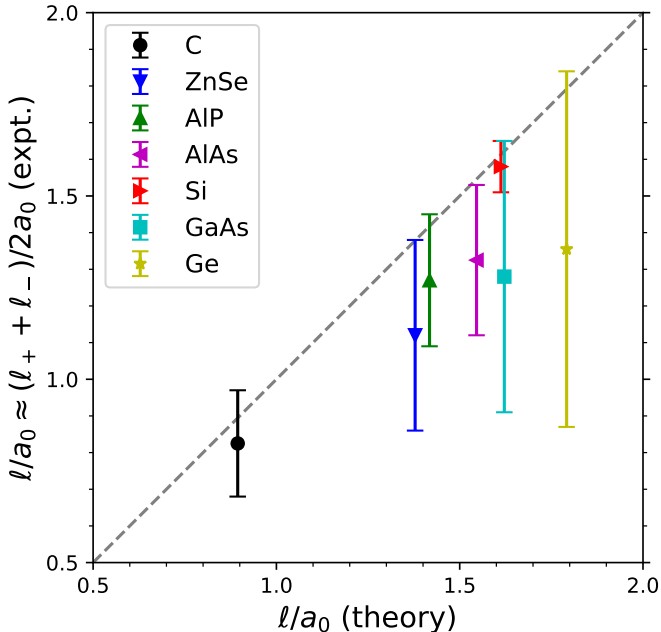

Figure 4: Comparison, for tetrehedrally-cordinated materials, between the $\ell$ values for valence electrons estimated from experimental data (Table 3), and those calculated from first principles in Ref. [10] using a pseudopotential method. The error bars indicate the range $[\ell_-, \ell_+]$.

## Acknowledgments

We are indebted to Morrel H. Cohen for an unpublished collaboration with one of us (I. S.), on which Problem 22.5 of Ref. [15] was based. We also wish to thank Liang Fu, Yugo Onishi, and Raffaele Resta for stimulating discussions. M. S. thanks the CCQ at the Flatiron Institute for hospitality while this research was carried out. The Flatiron Institute is a division of the Simons Foundation.

**Funding information**   Work by I. S. was supported by Grant No. PID2021-129035NB-I00 funded by MCIN/AEI/10.13039/501100011033 and by ERDF/EU. Work by M. S. was supported by the State Investigation Agency through the Severo Ochoa Programme for Centres of Excellence in R&D (CEX2023-001263-S), and from Generalitat de Catalunya (Grant No. 2021 SGR 01519).

## A   Polarization and localization in band insulators

In Eqs. (3) and (4), the electronic polarization and the electron localization tensor were written down for a generic bulk insulator (possibly correlated and/or disordered) using Kohn's center-of-mass operator. Alternatively, those expressions can be recast in terms of the Berry phase and quantum metric defined by the change in the many-body ground state under twisted boundary conditons [5, 40]. Here we specialize to the single-particle picture, and review the corresponding formulas for uncorrelated crystalline insulators.

The electronic polarization of a band insulator takes the form of a Berry phase of the cell-periodic Bloch states in momentum space [3,4],

$$\mathbf{P}_e = \frac{-|e|}{(2\pi)^3} \int d^3k \sum_{n=1}^{J} \mathbf{A}_{nn}(\mathbf{k}); \tag{A.1}$$

here $\mathbf{A}_{mn}(\mathbf{k}) = i\langle u_{m\mathbf{k}}|\nabla_{\mathbf{k}} u_{n\mathbf{k}}\rangle$ is the Berry connection matrix, the integral is over the first Brillouin zone (BZ), and the summation is over the valence bands. Alternatively, $\mathbf{P}_e$ can be written as [3,4]

$$\mathbf{P}_e = \frac{-|e|}{\Omega} \sum_{n=1}^{J} \langle \mathbf{r}\rangle_n, \tag{A.2}$$

where $\langle \mathbf{r}\rangle_n$ is the center of charge of a Wannier function constructed for band $n$.

The localization tensor can be obtained from the fluctuation-dissipation relation in Eq. (6). Using the Kubo-Greenwood formula for the optical conductivity, one finds the sum rule [5,8]

$$\int_0^\infty d\omega\, \omega^{-1} \operatorname{Re} \sigma_{ab}^S(\omega) = \frac{\pi e^2}{(2\pi)^3 \hbar} \int d^3k \sum_{n=1}^{J} g_{ab,nn}(\mathbf{k}). \tag{A.3}$$

On the right-hand side, $g(\mathbf{k})$ is the quantum metric tensor [6] of the valence manifold [4,9],

$$g_{ab,mn}(\mathbf{k}) = \frac{1}{2}\langle \partial_a u_{m\mathbf{k}}|Q_{\mathbf{k}}|\partial_b u_{n\mathbf{k}}\rangle + \frac{1}{2}\langle \partial_b u_{m\mathbf{k}}|Q_{\mathbf{k}}|\partial_a u_{n\mathbf{k}}\rangle, \tag{A.4}$$

with $\partial_a = \partial/\partial k_a$ and $Q_{\mathbf{k}} = \mathbb{1} - \sum_{n=1}^{J} |u_{n\mathbf{k}}\rangle\langle u_{n\mathbf{k}}|$. Inserting Eq. (A.3) in Eq. (6) gives

$$\ell_{ab}^2 = \frac{1}{(2\pi)^3 n_e} \int d^3k \sum_{n=1}^{J} g_{ab,nn}(\mathbf{k}), \tag{A.5}$$

which expresses the bulk localization tensor as a ground-state quantity.

For a one-dimensional (1D) insulator the localization tensor reduces to a scalar, and Eq. (A.5) can be written in terms of maximally-localized Wannier functions as

$$\ell^2 = \frac{1}{J} \sum_{n=1}^{J} \left( \langle x^2\rangle_n - \langle x\rangle_n^2 \right), \tag{A.6}$$

which follows from the relation between the BZ integral of the metric and the quadratic Wannier spread [9]. Thus, in 1D the localization tensor is equal to the average spread of the maximally-localized Wannier functions. More generally, in $d$ dimensions its Cartesian trace equals the gauge-invariant part of the average Wannier spread, which for $d > 1$ is smaller than the actual spread in any gauge [9].

In summary, electronic polarization is related to the Wannier centers of the valence bands, and the electron localization length squared (polarization fluctuations) gives a lower bound to the average Wannier spread.

# B  Longitudinal optical bounds

Here, we outline the extension to long-wave longitudinal modes [17,28] of the analysis carried out in Secs. 2 and 3 for transverse modes. We again assume cubic symmetry or higher so that $\epsilon_{ab}(\omega) = \delta_{ab}\epsilon(\omega)$, and define the moments of the energy-loss spectrum as

$$M_p = \frac{2}{\pi} \int_0^\infty d\omega\, \omega^p \operatorname{Im}\left[-\epsilon^{-1}(\omega)\right]. \tag{B.1}$$

The moments with $p = 1, 0, -1$ satisfy the relations

$$M_1 = \omega_{\mathrm{p}}^2, \tag{B.2a}$$

$$M_0 = \frac{2e^2}{\hbar \epsilon_0} n_{\mathrm{e}} \tilde{\ell}^2, \tag{B.2b}$$

$$M_{-1} = 1 - \epsilon^{-1}, \tag{B.2c}$$

where $\epsilon^{-1}$ stands for $\epsilon^{-1}(0)$. These are respectively the longitudinal counterpart of the oscillator-strength sum rule (10a) [21], the longitudinal fluctuation-dissipation relation (7), and the longitudinal counterpart of the Kramers-Krönig relation (10c).

Next, we introduce average gaps for longitudinal excitations by analogy with Eqs. (11-13),

$$\tilde{E}_{\mathrm{L}} = \frac{\hbar M_1}{M_0}, \qquad \tilde{E}_{\mathrm{P}}^2 = \frac{\hbar M_1}{\hbar^{-1} M_{-1}}, \tag{B.3}$$

$$\tilde{E}_{\mathrm{L}} = \frac{\hbar^2}{2m_{\mathrm{e}} \tilde{\ell}^2} \Leftrightarrow \left(\frac{\tilde{\ell}}{a_0}\right)^2 = \frac{\mathrm{Ry}}{\tilde{E}_{\mathrm{L}}}, \tag{B.4}$$

$$1 - \epsilon^{-1} = \left(\frac{\hbar \omega_{\mathrm{p}}}{\tilde{E}_{\mathrm{P}}}\right)^2. \tag{B.5}$$

Since the loss function appearing in Eq. (B.1) is positive semidefinite, one can immediately write down inequalities analogous to those in Eqs. (15), (16), and (18),

$$M_{p-q} \leq \frac{\hbar}{\tilde{E}_{\mathrm{G}}} M_{p-q+1} \leq \ldots \leq \left(\frac{\hbar}{\tilde{E}_{\mathrm{G}}}\right)^q M_p \tag{B.6}$$

($\tilde{E}_{\mathrm{G}}$ is the plasmon gap),

$$M_p^2 \leq M_{p-1} M_{p+1}, \tag{B.7}$$

and

$$\tilde{E}_{\mathrm{L}} \geq \tilde{E}_{\mathrm{p}} \geq \tilde{E}_{\mathrm{G}}, \qquad \tilde{E}_{\mathrm{P}}^2 \geq \tilde{E}_{\mathrm{G}} \tilde{E}_{\mathrm{L}}. \tag{B.8}$$

Finally, by forming the chained inequalities

$$\tilde{E}_{\mathrm{P}}^2 / \tilde{E}_{\mathrm{G}} \geq \tilde{E}_{\mathrm{L}} \geq \tilde{E}_{\mathrm{P}} \geq \tilde{E}_{\mathrm{G}}, \tag{B.9a}$$

$$\tilde{E}_{\mathrm{L}}^2 \geq \tilde{E}_{\mathrm{P}}^2 \geq \tilde{E}_{\mathrm{G}} \tilde{E}_{\mathrm{L}} \geq \tilde{E}_{\mathrm{G}}^2, \tag{B.9b}$$

$$\tilde{E}_{\mathrm{G}} \leq \tilde{E}_{\mathrm{P}}^2 / \tilde{E}_{\mathrm{L}} \leq \tilde{E}_{\mathrm{P}} \leq \tilde{E}_{\mathrm{L}}, \tag{B.9c}$$

and combining them with Eqs. (B.4) and (B.5), we obtain weak and strong bounds on $\tilde{\ell}^2$, $1 - \epsilon^{-1}$ and $\tilde{E}_{\mathrm{G}}$, respectively. Those bounds are given by Eqs. (19) and (20), with the replacements indicated in Eq. (21).

## C  Zero-point energy of the van der Waals crystal model

In this appendix we return to the van der Waals crystal model of Sec. 4.4, and calculate its zero-point energy in two different ways. First we use a Brillouin-zone integral,

$$E = \frac{\hbar}{2} \frac{\Omega}{(2\pi)^3} \int d^3q \sum_i \omega_i(\mathbf{q}). \tag{C.1}$$

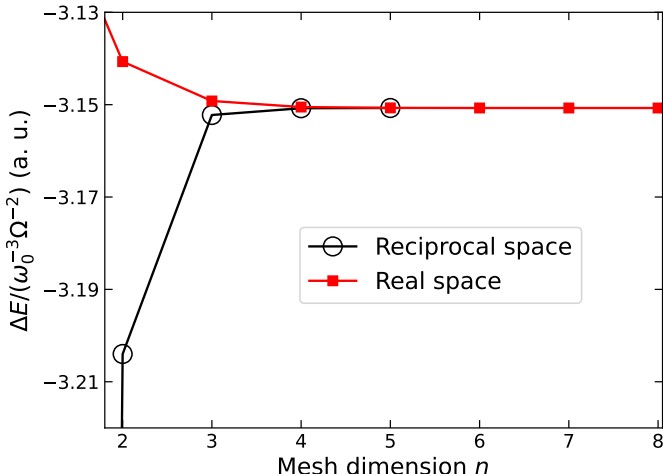

Figure 5: Convergence of the reciprocal- and real-space sums for the dispersion interaction energy using meshes of dimension $2n$. The plotted values correspond to Eqs. (C.4) and (C.6) for a simple-cubic lattice, in units of $\omega_0^{-3}\Omega^{-2}$ using Hartree atomic units (a. u.).

To verify that the normalization factors are correct, note that in the absence of interactions we recover the correct result for the isolated 3D oscillator,

$$E_0 = \frac{3}{2}\hbar\omega_0. \tag{C.2}$$

The interaction is regarded as a small perturbation, so we can Taylor-expand the square root of Eq. (42) for $\omega_i^2(\mathbf{q})$,

$$\omega_i = \sqrt{\omega_0^2 + \lambda_i} \simeq \omega_0 + \frac{1}{2}\frac{\lambda_i}{\omega_0} - \frac{1}{8}\frac{\lambda_i^2}{\omega_0^3}. \tag{C.3}$$

As the $C(\mathbf{q})$ matrix defined by Eq. (40) is traceless for all $\mathbf{q}$, the second term above drops out from Eq. (C.1). The leading correction is then given by the third term,

$$\Delta E = -\frac{\hbar}{16\omega_0^3}\frac{\Omega}{(2\pi)^3}\int d^3q \sum_i \lambda_i^2(\mathbf{q}). \tag{C.4}$$

Overall, the interaction energy is negative and in view of Eq. (43) it appears to scale as $\Omega^{-2}$, which at first sight seems consistent with van der Waals. This is confirmed by a numerical evaluation of Eq. (C.4) for a simple-cubic lattice (Fig. 5), which shows a $\Omega^{-2}$ behavior for $\Delta E$ in the limit of a dense $\mathbf{q}$ mesh.

As further validation, we have computed the same energy as a real-space sum of pair interactions. We start from the interaction energy of the 3D dimer model of Sec. 4.3, which is obtained by expanding Eqs. (32) and (33) according to Eq. (C.3). The result [31]

$$\Delta E_{12} = -\frac{3}{4}\left(\frac{\alpha_0'}{R^3}\right)^2 \hbar\omega_0, \tag{C.5}$$

which is enhanced by a factor of 3/2 relative to that of the 1D dimer model [34], leads to a crystal energy of

$$\Delta E = -\hbar\left(\frac{e^2/m_e}{4\pi\epsilon_0}\right)^2 \frac{3}{8\omega_0^3}\sum_{\mathbf{R}\neq\mathbf{0}}\frac{1}{R^6}. \tag{C.6}$$



(Note the additional factor of 1/2 to avoid double counting of the pair interactions.) As shown in Fig. 5, the converged value of this real-space summation agrees with that of the reciprocal-space summation (C.4). The plotted quantity is $\Delta E/(\omega_0^{-3}\Omega^{-2})$ in Hartee atomic units, and its converged value is precisely $-(3/8)A_6$, where

$$A_6 \equiv \sum_{i,j,k}'(i^2 + j^2 + k^2)^{-3} \simeq 8.40192 \tag{C.7}$$

(with $i = j = k = 0$ excluded), is a lattice sum tabulated by Lennard-Jones and Ingham [41].

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
