# Peer review of "Optical bounds on many-electron localization"

_SciPost Physics, doi:SciPost Phys. 18, 127 (2025)_

## Round 1 · Referee Report · Anonymous (Referee 1) · 2025-1-2

Strengths

The paper is topical. There is lots of recent work investigating the significance of various negative moment sum rules of the conductivity and their relation to quantum geometric effects. This paper works out some consequences of these considerations.

They establish inequalities between various parameters and length scales and connect them to optical sum rules, and organize them into weak and strong bounds on three characteristic properties of insulators: electron localization length l, electric susceptibility χ, and the optical gap EG.

The paper is interesting from a theoretical perspective as these bounds providing an organizing idea for understanding the physics. They may also be useful for experimentalists in interpreting their data.

Weaknesses

There are no real weaknesses per se, although I wonder if there is a even tighter bond lurking around as even the valence only strong bound still under constrains the localization length

Report

I think the paper should be accepted as is.

Recommendation

Publish (easily meets expectations and criteria for this Journal; among top 50%)

  • validity: top
  • significance: good
  • originality: high
  • clarity: top
  • formatting: excellent
  • grammar: excellent

Author:  Ivo Souza  on 2025-03-01  [id 5252]

(in reply to Report 2 on 2025-01-02)

One way to obtain tighter bounds is by invoking sum rules
containing moments of the absorption spectrum that are different from
the ones we have considered: I_0, I_1 and I_2, which correspond to
m_1, m_0, and m_{-1} in the notation of Traini (Ref. 19 of the revised
manuscript). In this way, Traini obtained improved estimates for the
atomic susceptibility: the estimate in his Eq. (39), which corresponds
to the lower and strong upper bounds in Eq. (25b) of the manuscript,
is improved by his Eq. (41), which involve the higher spectral moments
m_2 and m_3. It is unclear how to relate those higher moments to
tabulated properties of solids, which limits their practical utility.

We now mention in passing these tighter bounds in a sentence below
Eq. (25):

"Interestingly, both bounds can be improved by means of correction
terms involving positive moments of the absorption spectrum [19]."

---

## Round 1 · Referee Report · Anonymous (Referee 1) · 2025-1-2

Strengths

The paper is topical. There is lots of recent work investigating the significance of various negative moment sum rules of the conductivity and their relation to quantum geometric effects. This paper works out some consequences of these considerations.

They establish inequalities between various parameters and length scales and connect them to optical sum rules, and organize them into weak and strong bounds on three characteristic properties of insulators: electron localization length l, electric susceptibility χ, and the optical gap EG.

The paper is interesting from a theoretical perspective as these bounds providing an organizing idea for understanding the physics. They may also be useful for experimentalists in interpreting their data.

Weaknesses

There are no real weaknesses per se, although I wonder if there is a even tighter bond lurking around as even the valence only strong bound still under constrains the localization length

Report

I think the paper should be accepted as is.

Recommendation

Publish (easily meets expectations and criteria for this Journal; among top 50%)

  • validity: top
  • significance: good
  • originality: high
  • clarity: top
  • formatting: excellent
  • grammar: excellent

Author:  Ivo Souza  on 2025-02-28  [id 5250]

(in reply to Report 1 on 2025-01-02)

One way to obtain tighter bounds is by invoking sum rules
containing moments of the absorption spectrum that are different from
the ones we have considered: I_0, I_1 and I_2, which correspond to
m_1, m_0, and m_{-1} in the notation of Traini (Ref. 18 of the revised
manuscript). In this way, Traini obtained improved estimates for the
atomic susceptibility: the estimate in his Eq. (39), which corresponds
to the lower and strong upper bounds in Eq. (25b) of the manuscript,
is improved by his Eq. (41), which involve the higher spectral moments
m_2 and m_3. It is unclear how to relate those higher moments to
tabulated properties of solids, which limits their practical utility.

We now mention in passing these tighter bounds in a sentence below
Eq. (25):

"Interestingly, both bounds can be improved by means of correction
terms involving positive moments of the absorption spectrum [18]."

---

## Round 1 · Referee Report · Anonymous (Referee 2) · 2025-1-14

Report

In this work, the authors present a systematic and rigorous study of the electronic properties of insulators, focusing on their behavior as described by the optical sum rule. The sum-rule inequalities reveal the relationship among the localization gap (EL), the Penn gap (Ep), and the optical gap (EG), which serve as the foundation for determining lower and upper bounds of the localization length (l). In the second part of the study, single-electron and two-electron (van der Waals dimer model) systems are solved analytically, demonstrating agreement with the inequalities and localization results. The localization lengths of various materials are estimated, showing reasonable consistency with data obtained from first-principles calculations.
I find this work well-written and clearly explained. While the localization length estimation relies on experimental data and may predict a relatively broad range compared to first-principles methods, I believe this work should be published without significant changes.

Requested changes

Questions (not necessary to request changes):

  1. As the Drude weight is introduced at the beginning of the paper, is it also possible to characterize it using optical absorption equations? In other words, can this methodology be extended to metallic systems without a fundamental gap in the DC limit?

  2. In Section 2, paragraph 2, there is an assumption: "For simplicity, we will assume cubic symmetry or higher so that...". I wonder if this assumption is overly simplified when considering the localization length in real materials. Would the results hold for systems with lower symmetry?

  3. The Van der Waals dimer model, where the two-electron system is analyzed in detail, provides a good starting point for studying many-body physics. However, since the correlation function is not considered and the resulting localization length does not differ significantly, I am curious about the rationale for introducing this many-body model. Does it offer additional insights into specific material properties, especially given that the results seem to saturate in the long-wavelength limit?

  4. In the comparative analysis shown in Figure 4, only tetrahedrally-coordinated materials are included. Is it possible to extend the comparison to other material systems mentioned in the article, such as diamond, zincblende, or rocksalt alkali halides? Furthermore, do the differences between first-principles and the presented method depend on crystal symmetry? For example, do tetrahedrally-coordinated materials exhibit better agreement due to their higher symmetry?

  5. Following the previous question, if EG represents both the band structure energy gap and the optical gap (using the same notation), is it possible to extract EL and Ep directly from the band structure? If so, would experimental bounds no longer be necessary?

Attachment

Recommendation

Publish (easily meets expectations and criteria for this Journal; among top 50%)

  • validity: high
  • significance: good
  • originality: high
  • clarity: top
  • formatting: excellent
  • grammar: perfect

Author:  Ivo Souza  on 2025-03-01  [id 5253]

(in reply to Report 3 on 2025-01-14)

R1: Our focus in the manuscript was on insulators, where optical
absorption is pureley interband and the I_1 and I_2 spectral moments
are finite (they diverge in conductors due to intraband
absorption). However, the sum-rule approach to obtain optical bounds
only relies on the positive semidefiniteness of the absorptive optical
conductivity, a property that is satisfied by the interband and
intraband (Drude) absorption tensors separately. It may therefore be
possible to obtain optical bounds on the Drude weight (the intraband
I_0 spectral moment) in terms of positive spectral moments of the
intraband spectrum. This could be a topic for future investigations.

R2: The assumption of cubic symmetry is not essential. It was only
made for the sake of notational simplicity, and because the materials
considered are all cubic. In the general case, the optical absorption
of linearly-polarized light is described by a symmetric conductivity
tensor, and the sum rules also yield symmetric tensors. In particular,
the anisotropic localization properties are described by a
localization tensor, whose eigenvalues are the localization lengths
squared along its principal axes. With these tensorial generalization,
all the results discussed in the manuscript can be readily extended to
systems with lower symmetry. For example, in Ref. [18] of the revised
manuscript the in-plane electron localization was studied via optical
bounds for materials with C_3 or C_4 rotational symmetry.

The following sentence was added to the Conclusions:

"The extension to low-symmetry crystals with anisotropic localization
and susceptibility tensors is also straightforward."

R3: Part of the motivation for studying the van der Walls dimer model
(and its crystal generalization) was indeed as a starting point to
understand how electron correlations affect the polarization
fluctuations. The role of long-ranged electrostatics was emphasized by
Resta in Ref. 13, and in the manuscripted we showed how the impact of
dispersion forces on the optical bounds can be illustrated by means of
an exactly-solvable model of coupled oscillators. Although the model
is probably too simple to provide insights into the properties of
specific materials, it allows to draw some general conclusions. In
particular, as discussed at the end of Sec. 4.3, it illustrates how
the fluctuation-dissipation sum rule fails if the electron-electron
interaction is treated at the mean-field level, e.g., within
Hartree-Fock theory. Regarding the saturation of the bounds in this
model, it follows from the dipole selection rules for the harmonic
oscillator. For an anharmonic oscillator, the bounds would no longer
be saturated.

R4: For our comparative analysis in Fig. 4, we relied on the
first-principles results of Sgiarovello et al (Ref. 10), which only
included tetrahedrally-coordinated semiconductors with the diamond and
zincblende structures. The analysis is however quite general, and
should be applicable to other materials with different types of
crystals structures, including lower-symmetry ones. We agree that it
would be worthwhile to carry out in the future a more systematic
study, where the localization tensor is calculated from first
principles for a wide range of materials with different symmetries.

R5:The localization gap EL and the Penn gap Ep are not "gaps" in the
strict sense of the word. Rather, they are transition energies
weighted by the absorption strength ("average gaps"); as such, they
cannot be extracted directly from the band structure (the eigenvalues
as a function of momentum) in the same way as the minimum direct gap
EG can. This is precisely what makes the experimental bounds useful,
as they allow to estimate the value of a non-trivial ground-state
quantity (the localization length) which is not a simple bandstructure
property.

To clarify this point, we have rephrased the sentence below Eq. (11):

"These average inverse excitation energies weighted by the transition
strength [19] will be denoted as (inverse) 'average gaps.'"

Author:  Ivo Souza  on 2025-02-28  [id 5251]

(in reply to Report 3 on 2025-01-14)

R1: Our focus in the manuscript was on insulators, where optical
absorption is pureley interband and the I_1 and I_2 spectral moments
are finite (they diverge in conductors due to intraband
absorption). However, the sum-rule approach to obtain optical bounds
only relies on the positive semidefiniteness of the absorptive optical
conductivity, a property that is satisfied by the interband and
intraband (Drude) absorption tensors separately. It may therefore be
possible to obtain optical bounds on the Drude weight (the intraband
I_0 spectral moment) in terms of positive spectral moments of the
intraband spectrum. This could be a topic for future investigations.

R2: The assumption of cubic symmetry is not essential. It was only
made for the sake of notational simplicity, and because the materials
considered are all cubic. In the general case, the optical absorption
of linearly-polarized light is described by a symmetric conductivity
tensor, and the sum rules also yield symmetric tensors. In particular,
the anisotropic localization properties are described by a
localization tensor, whose eigenvalues are the localization lengths
squared along its principal axes. With these tensorial generalization,
all the results discussed in the manuscript can be readily extended to
systems with lower symmetry. For example, in Ref. [19] of the revised
manuscript the in-plane electron localization was studied via optical
bounds for materials with C_3 or C_4 rotational symmetry.

The following sentence was added to the Conclusions:

"The extension to low-symmetry crystals with anisotropic localization
and susceptibility tensors is also straightforward."

R3: Part of the motivation for studying the van der Walls dimer model
(and its crystal generalization) was indeed as a starting point to
understand how electron correlations affect the polarization
fluctuations. The role of long-ranged electrostatics was emphasized by
Resta in Ref. 14, and in the manuscripted we showed how the impact of
dispersion forces on the optical bounds can be illustrated by means of
an exactly-solvable model of coupled oscillators. Although the model
is probably too simple to provide insights into the properties of
specific materials, it allows to draw some general conclusions. In
particular, as discussed at the end of Sec. 4.3, it illustrates how
the fluctuation-dissipation sum rule fails if the electron-electron
interaction is treated at the mean-field level, e.g., within
Hartree-Fock theory. Regarding the saturation of the bounds in this
model, it follows from the dipole selection rules for the harmonic
oscillator. For an anharmonic oscillator, the bounds would no longer
be saturated.

R4: For our comparative analysis in Fig. 4, we relied on the
first-principles results of Sgiarovello et al (Ref. 10), which only
included tetrahedrally-coordinated semiconductors with the diamond and
zincblende structures. The analysis is however quite general, and
should be applicable to other materials with different types of
crystals structures, including lower-symmetry ones. We agree that it
would be worthwhile to carry out in the future a more systematic
study, where the localization tensor is calculated from first
principles for a wide range of materials with different symmetries.

R5:The localization gap EL and the Penn gap Ep are not "gaps" in the
strict sense of the word. Rather, they are transition energies
weighted by the absorption strength ("average gaps"); as such, they
cannot be extracted directly from the band structure (the eigenvalues
as a function of momentum) in the same way as the minimum direct gap
EG can. This is precisely what makes the experimental bounds useful,
as they allow to estimate the value of a non-trivial ground-state
quantity (the localization length) which is not a simple bandstructure
property.

To clarify this point, we have rephrased the sentence below Eq. (11):

"These average inverse excitation energies weighted by the transition
strength [18] will be denoted as (inverse) 'average gaps.'"

---

## Round 2 · Referee Report · Anonymous (Referee 2) · 2025-3-9

Report

The author has answered all the questions point by point and revised all the necessary parts in the main text.

Recommendation

Publish (easily meets expectations and criteria for this Journal; among top 50%)

---

## Round 2 · Referee Report · Anonymous (Referee 1) · 2025-3-20

Strengths

the authors have responded to my comments. I think the paper should be published.

Weaknesses

no weaknesses

Report

Publish as is

Recommendation

Publish (easily meets expectations and criteria for this Journal; among top 50%)

---

## Round 2 · Author Response

Dear Editors,

We would like to thank both referees for their thoughtful and constructive remarks on our manuscript. As requested, we have made minor changes in response to their comments and suggestions. With the goal of further improving the text, we made a few additional changes at various places, and corrected some typos.

---

## Round 2 · List of Changes

Changes in response to the comments of the referees:

  • We have rephrased the sentence below Eq. (11): "These average inverse excitation energies weighted by the transition strength [18] will be denoted as (inverse) 'average gaps.'"

  • A sentence was added below Eq. (25): "Interestingly, both bounds can be improved by means of correction terms involving positive moments of the absorption spectrum [18]."

  • A sentence was added in the Conclusions: "The extension to low-symmetry crystals with anisotropic localization and susceptibility tensors is also straightforward."

Additional changes not directly motivated by the referee reports:

  • The notation for the localization tensor in Eqs. (4-7) has changed, and the meaning of Eq. (5) has also changed slightly.

  • Equation (7) has been corrected. The longitudinal localization length squared was previously written as a Cartesian tensor, when it is in fact a direction-dependent scalar.

  • The caption of Table 1 was shortened.

  • At the end of the paragraph below Eq. (10), a discussion was added on the application of optical sum rules to characterize F centers in alkali halide crystals, with a new reference [24]. References [22,23] were also added in the same paragraph, in connection with atomic sum rules.

  • The paragraph about Chern insulators at the end of Sec. 3 was shortened slightly, by removing a sentence about time-odd sum rules and inequalities that fall outside the scope of the present work.

  • Three footnotes were removed.

  • Other minor edits were made to improve the clarity of the text.

---

## Editorial Decision

published